# An Integrated In Silico and In Vitro Approach for the Identification of Natural Products Active against SARS-CoV-2

**DOI:** 10.3390/biom14010043

**Published:** 2023-12-28

**Authors:** Rosamaria Pennisi, Davide Gentile, Antonio Rescifina, Edoardo Napoli, Paola Trischitta, Anna Piperno, Maria Teresa Sciortino

**Affiliations:** 1Department of Chemical, Biological, Pharmaceutical and Environmental Science, University of Messina, Viale Ferdinando Stagno d’Alcontres 31, 98166 Messina, Italy; paola.trischitta@studenti.unime.it (P.T.); anna.piperno@unime.it (A.P.); mtsciortino@unime.it (M.T.S.); 2Department of Chemistry, Materials and Chemical Engineering “G. Natta”, Politecnico di Milano, Via Mancinelli 7, 20131 Milano, Italy; 3Department of Drug and Health Sciences, University of Catania, V.le A. Doria 6, 95125 Catania, Italy; arescifina@unict.it; 4Istituto di Chimica Biomolecolare—Consiglio Nazionale delle Ricerche, 95126 Catania, Italy; edoardo.napoli@icb.cnr.it; 5Department of Chemistry, Biology and Biotechnology, University of Perugia, Via Elce di Sotto 8, 06123 Perugia, Italy

**Keywords:** SARS-CoV-2, natural antivirals, pseudoviruses, luteolin-7-*O*-glucuronide, folic acid rosmarinic acid, cynarin

## Abstract

Coronavirus disease 2019 (COVID-19), caused by severe acute respiratory syndrome coronavirus 2 (SARS-CoV-2), has provoked a global health crisis due to the absence of a specific therapeutic agent. 3CL^pro^ (also known as the main protease or M^pro^) and PL^pro^ are chymotrypsin-like proteases encoded by the SARS-CoV-2 genome, and play essential roles during the virus lifecycle. Therefore, they are recognized as a prospective therapeutic target in drug discovery against SARS-CoV-2 infection. Thus, this work aims to collectively present potential natural 3CL^pro^ and PL^pro^ inhibitors by in silico simulations and in vitro entry pseudotype-entry models. We screened luteolin-7-*O*-glucuronide (L7OG), cynarin (CY), folic acid (FA), and rosmarinic acid (RA) molecules against PL^pro^ and 3CL^pro^ through a luminogenic substrate assay. We only reported moderate inhibitory activity on the recombinant 3CL^pro^ and PL^pro^ by L7OG and FA. Afterward, the entry inhibitory activity of L7OG and FA was tested in cell lines transduced with the two different SARS-CoV-2 pseudotypes harboring alpha (α) and omicron (o) spike (S) protein. The results showed that both compounds have a consistent inhibitory activity on the entry for both variants. However, L7OG showed a greater degree of entry inhibition against α-SARS-CoV-2. Molecular modeling studies were used to determine the inhibitory mechanism of the candidate molecules by focusing on their interactions with residues recognized by the protease active site and receptor-binding domain (RBD) of spike SARS-CoV-2. This work allowed us to identify the binding sites of FA and L7OG within the RBD domain in the alpha and omicron variants, demonstrating how FA is active in both variants. We have confidence that future in vivo studies testing the safety and effectiveness of these natural compounds are warranted, given that they are effective against a variant of concerns.

## 1. Introduction

In the last few decades, the recurring occurrences of both epidemic and pandemic viruses worldwide, i.e., those caused by SARS-CoV-1 (2002), Middle East respiratory syndrome coronavirus (MERS-CoV) (2012), and SARS-CoV-2 (2019), have indicated that viral outbreaks of emerging viruses are becoming a severe threat for the public health. Despite the accumulation of a large body of knowledge about many human viral pathogens’ replicative and pathogenetic mechanisms, approved antiviral drugs prior to COVID-19 were limited only to a confined array of viruses (e.g., HIV, HBV, HCV, some Herpesviruses, Influenza virus). The great efforts to find therapeutic solutions against SARS-CoV-2 have led to identifying several functional proteins that can be targeted with small-molecule inhibitors. Among these, two viable main viral proteases, namely 3CL^pro^ and PL^pro^, and some cellular proteases involved in the complex mechanisms of SARS-CoV-2 entry into host cells, were identified as valuable molecular targets due to their crucial role in the virus replication cycle [1,2,3]. Recently, the first oral drug targeting 3CL^pro^, namely Nirmatrelvir, has been marketed by Pfizer for the treatment of severe forms of COVID-19 [4,5]. Learning from this recent success, and considering the emergence of new variants (i.e., alpha, beta, gamma, delta, and omicron), capable of adapting better to the host, being transmitted more effectively, evading the immune defenses, and potentially being resistant to therapeutic medicines, new strategies able to reduce the risk of drug resistance associated with viral mutations have become imperative. In this scenario, an exciting and challenging approach consists of target multi-mechanisms of action of some inhibitors capable of targeting viral 3CL^pro^/PL^pro^ as well as host cellular proteases such as furin-like proteases, transmembrane serine protease 2 (TMPRSS2), and cathepsin L (CatL) [6,7]. While TMPRSS2 and cathepsins contribute to determining the entry of SARS-CoV-2, the viral proteases 3CL^pro^/PL^pro^ are essential for replication and generating mature viral proteins [8]. Indeed, SARS-CoV-2 entry into the host cell requires priming of the S protein by cleavage at the S1/S2. This cleavage is carried out primarily by the serine protease TMPRSS2, but can also be performed by the cysteine protease CatL [9,10,11]. Based on this knowledge, the proteases represent an ideal antiviral therapeutic target [12]. Taking into account the growing number of studies related to the beneficial effects of plant-derived small organic compounds [13,14] to influence viral replication [15,16], we selected some representative natural compounds endowed with relevant biological properties and their antiviral properties, investigated in depth with an integrated in silico and in vitro approach to identify anti-SARS-CoV-2 agents that are effective on mutated viruses. Specifically, we focused on L7OG, CY, FA, and RA. L7OG is the natural glycoside of luteolin, an essential dietary flavonoid present in different plant species [17]. Recently, the interactions of luteolin with the viral S-protein RBD-reducing angiotensin-converting enzyme (ACE2) were described [18]. CY (1,3-dicaffeoylquinic acid) is a natural polyphenolic compound with relevant biological activities, including antioxidant, antiviral, immunomodulatory, and vasodilator functions [19]. Although the protective effects against SARS-CoV-2 infection were described for some plant extracts containing CY, the role of polyphenols and viral targets remains unclear [20]. FA, named vitamin B9, was included in our study since in silico studies and molecular docking supported its action as an inhibitor for the SARS-CoV-2 virus entry into host cells and viral replication [21]. However, the action of FA in SARS-CoV-2 viral infection is still a matter of debate. RA, the ester of caffeic acid with the α-hydroxyl of 3-(3′,4′-dihydroxyphenyl)lactic acid, was described as a SARS-CoV-2 M^pro^ inhibitor at a micromolar concentration (IC_50_ = 6.84 μM) [22]. Exciting results have been reported from in silico medicinal chemistry, which has attracted significant research interest, particularly in drug reuse, reducing the time and costs of developing new pharmaceutical products [14,23,24]. In the fight against COVID-19, the effectiveness of artificial intelligence approaches has also been demonstrated with the more classic computational studies based on ligands and structure [14,23,24].

In line with this research topic, we employed an integrated in silico/in vitro approach to discover natural products active against SARS-CoV-2. From an initial screening of the inhibition of viral proteases, performed using 300 μM of compounds, we obtained that L7OG and FA significantly reduced the % of SARS-CoV-2 3CL^pro^ and PL^pro^ activity. CY and RA, however, exhibited a non-significant reduction. For this reason, we focused on evaluating the activity of L7OG and FA against SARS-CoV-2 through pseudovirus assay and in silico analysis. The results obtained from our parallel in vitro and in silico simulation studies of L7OG and FA identified the binding sites of FA and L7OG within the RBD domain in the alpha and omicron variants, and suggest the ability of FA to interfere with the entry machinery of both variants by acting in different RBD domain regions.

The spike protein encoded by the virus, which forms trimer spikes on the surface of the virus, allows for entry of SARS-CoV-2 into host cells. The RBD domain of the S trimer accommodates two distinct “up” and “down” conformations, of which only the “up” RBDs bind the human host receptor protein ACE2. Some compounds, such as fatty acids (e.g., linoleic acid) or retinol derivatives, can bind to a hydrophobic pocket of the RBD domain, stabilizing the locked S conformation and reducing interaction with ACE2. The hallmarks of free fatty acid (FFA) binding pockets in the S protein are an extended “fatty” tube, surrounded by hydrophobic amino acids, which accommodates the lipophilic hydrocarbon tail; and a hydrophilic anchor, often positively charged, to bind—via electrostatic interactions and salt bridges—the acid head of FFA [25,26].

The workflow of the designed strategy (Figure 1) included (i) identification of 3CL^pro^ or PL^pro^ inhibitors by enzymatic biological assays; (ii) evaluation of the ability to inhibit the virus entry using MLV-based SARS-CoV-2 pseudotype, α-SARS-CoV-2, and o-SARS-CoV-2 variants; (iii) the study of dynamic interaction with the pseudotype particles. The MLV-based SARS-CoV-2 pseudotypes, α-SARS-CoV-2 and o−SARS-CoV-2 variants, were employed to simulate the infection and identify natural products active against SARS-CoV-2. The pseudovirus technology for the production of both variants fulfills different needs of working and research efficiency: (i) it represents a safe alternative to evaluate potential entry inhibitor antivirals in a BSL-2 environment instead of Biosafety Level 3 (BSL-3) laboratories required for highly infective SARS-CoV-2 [27,28,29]; (ii) it allows to use different pseudotyped variants to elucidate some mechanism underlying the interactions between virus and cellular membrane receptors; (iii) it enables the application of different experiment approaches, i.e., cell pretreatment and virus pretreatment, to discriminate whether the screened antiviral compounds act against the cellular or viral receptors.

## 2. Materials and Methods

### 2.1. Cells

VERO cell lines (American Type Culture Collection) were propagated in minimal essential medium (EMEM) and supplemented with 6% fetal bovine serum (FBS) (Lonza, Basel, Switzerland) at 37 °C under 5% CO_2_.

### 2.2. Materials

L7OG was purchased by HWI Pharmascience GmbH (Ruelzheim, Germany). CY, RA, and FA were purchased by Sigma (Merck, Darmstadt, Germany) and used to prepare the respective stock solution.

### 2.3. SARS-CoV-2 3CL^pro^ and PL^pro^ Luminescent Assays

The inhibitory activities of the natural compounds against 3CL^pro^ and PL^pro^ were determined using the SARS-CoV-2 3CL^pro^ and PL^pro^ luminescent assays as reported by manufacturer instructions. The purified recombinant SARS-CoV-2 3CL^pro^ (16 μg/mL) (BPS Bioscience cat# 100823) and 10 nM of purified GST-PL^pro^ (R&D Systems, Cat# E-611-050) were separately combined with a 12.5 μL of 4X of L7OG, RA, CY, FA, diluted in assay buffer (50 mM HEPES pH 7.2, 10 mM DTT, and 0.1 mM EDTA) and added to 3CL^pro^ and PL^pro^ substrate solutions (40 μM) in an opaque white 96-well plate for 60 min at 37 °C. GC376 was used as a positive control. Reactions were then terminated by adding 50 μL of Luciferin Detection Reagent (Promega cat# V8920/V8921), and after 20 min at room temperature, luminescence was recorded on a GloMax^®^ luminometer.

### 2.4. Viability Assay

A CCK-8 assay (ab228554; Abcam, Cambridge, UK) was performed to evaluate the cytotoxicity of pure compounds. WST-8/CCK8 tetrazolium salt is reduced by cellular dehydrogenases to form an orange formazan product that is soluble in a tissue culture medium. The formazan produced is directly proportional to the number of living and metabolically active cells and is measured via absorbance at 460 nm [30]. Therefore, VERO cells (2 × 10^4^ cells/mL) were grown in 96-well microtiter plates at 37 °C in a 5% CO_2_ incubator for 24 h. Then, they were exposed to serial dilutions of L7OG CY, FA, and RA for 72 h and incubated with CCK8 tetrazolium salt for 4 h at 37 °C in a CO_2_ incubator. The absorbance was measured at 460 nm using a GloMax^®^ Discover Microplate Reader (Promega, Madison, WI, USA), and the percentage of cellular viability was calculated and compared to untreated cells.

### 2.5. Production of SARS-CoV-2 Pseudotyped Particles

For the study of SARS-CoV-2 infection in a BSL-2 laboratory, a SARS-CoV-2 pseudovirus particle production and infection system was constructed using a lentiviral vector-bearing luciferase gene reporter for easy observation and analysis. The protocol to generate SARS-CoV-2 pseudoviruses consisted of a three-plasmid co-transfection strategy in HEK-293T cells and was previously reported [31]. pcDNA3.1(-) SARS-Swt-C9 plasmid for α-SARS-CoV-2 variant was gently provided by Jean K. Millet [32], and o−-SARS-CoV-2 variant (BA.2 lineage) plasmid was purchased by Invivogen (Catalog code: p1-spike-v12). Pseudovirions were produced using Lipofectamine by co-transfection of 300 ng pCMV-MLVgag-pol, 400 ng pTG-Luc, and 300 ng spike encoding vector. During the production of pseudotyped particles, negative and positive controls were produced by replacing the plasmid encoding the spike glycoprotein with an empty vector and VSV envelope, respectively. The production efficiency of the pseudoviruses was assessed by measuring the luciferase gene expression with the Luciferase Assay System (Promega) according to the manufacturer’s instructions.

### 2.6. In Vitro Pseudovirus Entry Inhibition Assay

For pseudovirus entry inhibition screening, 4 × 10^4^ VERO cells were seeded in a 96-well plate in 100 μL of DMEM medium containing 6% of FBS and pretreated with 20 μM concentration of FA or 5 μM concentration of L7OG for 2 h, followed by transduction with α and o-SARS-CoV-2 spike pseudotyped virus particles. In addition, to investigate the inhibition effect on pseudovirus infection by L7OG and FA, α and ο−-SARS-CoV-2 spike pseudotyped virus particles were pretreated with 20 μM concentration of FA and 5 μM concentration of L7OG for 1 h at 4 °C. 72 h post-transduction, the cells were lysed, and the luciferase activity was determined. Briefly, 50 μL of the cell lysis buffer reagent (Promega Inc.) was added to each well after removing media, and plates were incubated with shaking for 10 min; 100 μL of luciferin substrate was added to each well, and luminescence was read with 1 min integration and delay time. The assay was performed in triplicate, and the data were reported as relative light units (RLUs) compared to uninfected control. The positive controls Calpeptin (2 μM) (Bio-Techne SRL, Catalog # 0448) and Camostat (100 μM) were used in each plate as CatL and TMPRSS2 inhibitors, respectively.

### 2.7. Molecular Docking

Flexible ligand docking experiments were performed by employing AutoDock 4.2.6 software implemented in YASARA (v. 23.5.19, YASARA Biosciences GmbH, Vienna, Austria), using the crystal structure of SARS-CoV-2 3CL^pro^ (PDB ID: 6LU7), SARS-CoV-2 PL^pro^ (PDB ID: 6W9C), SARS-CoV-2 o−-RBD (PDB ID: 7WBL) complex with human ACE2, SARS-CoV-2 spike glycoprotein in complex with all-*trans* retinoic acid (PDB ID: 7Y42) and SARS-CoV-2 omicron S-close (PDB ID: 7WK2), obtained from the Protein Data Bank (PDB, http://www.rcsb.org/pdb) URL (accessed on 2 April 2023), and the Lamarckian genetic algorithm (LGA). The maps were generated by the program AutoGrid (4.2.6) with a spacing of 0.375 Å and dimensions that encompass all atoms extending 5 Å from the surface of the structure of the crystallized ligand. Point charges were initially assigned according to the AMBER03 force field and then damped to mimic the less polar Gasteiger charges used to optimize the AutoDock scoring function. The structure of all ligands was optimized at the semiempirical level of PM6 theory [33]. As previously reported, all parameters were inserted at their default settings [34]. In the docking tab, the macromolecule and ligand were selected, and GA parameters were set as ga_runs = 100, ga_pop_size = 150, ga_num_evals = 25,000,000, ga_num_generations = 27,000, ga_elitism = 1, ga_mutation_rate = 0.02, ga_crossover_rate = 0.8, ga_crossover_mode = two points, ga_cauchy_alpha = 0.0, ga_cauchy_beta = 1.0, number of generations for picking worst individual = 10.

### 2.8. Molecular Dynamics Simulations

The molecular dynamics simulations of the protein/ligand complexes were performed with the YASARA Structure package according to the previously reported procedures [35,36]. A periodic cubic simulation cell with boundaries extending 8 Å [37] from the surface of the complex was employed. The box was filled with water, with a maximum sum of all water bumps of 1.0 Å and a density of 0.997 g mL^−1^.

The setup included optimizing the hydrogen bonding network [38] to increase the solute stability and a p*K*_a_ prediction to fine-tune the protonation states of protein residues at the chosen pH of 7.4 [39] NaCl ions were added with a physiological concentration of 0.9%, with either Na or Cl excess to neutralize the cell. Water molecules were deleted to readjust the solvent density to 0.997 g/mL. The simulation was run using the ff14SB force field [40] for the solute, GAFF2 [41] and AM1BCC [42] for ligands, and TIP3P for water. The cutoff was 10 Å for van der Waals forces (the default used by AMBER [43]), and no cutoff was applied to electrostatic forces (using the particle mesh Ewald algorithm [44]). The equations of motions were integrated with multiple time steps of 2.5 fs for bonded interactions and 5.0 fs for nonbonded interactions at a temperature of 298 K and a pressure of 1 atm (NPT ensemble) using algorithms described in detail previously [45,46]. The final system dimensions were approximately 80 × 80 × 80 Å^3^. A short MD simulation was run on the solvent only to remove clashes. The entire system was then energy minimized using the steepest descent minimization to remove conformational stress, followed by a simulated annealing minimization until convergence (<0.01 kcal/mol Å). Finally, 100 ns of MD simulations without any restrictions were conducted, and the conformations of each system were recorded every 200 ps. After inspection of the solute RMSD as a function of simulation time, the last 3 ns averaged structures were considered for further analysis.

### 2.9. Statistical Analysis

Three independent experiments were carried out in triplicate (n = 3) for each assay, and the results represent the average ±SD. The statistical analysis was performed with GraphPad Prism 8 software (Graph-Pad Software, San Diego, CA, USA) using one-way variance analysis (ANOVA). The significance of the *p*-value is indicated with asterisks (*, **, ***, ****), denoting the *p*-value significance levels less than 0.05, 0.01, 0.001, and 0.0001, respectively. The half-maximal cytotoxic concentration (CC_50_) and half-maximal inhibitory concentration (IC_50_) were calculated using nonlinear regression analysis with Graph Pad Prism Version 8.

## 3. Results

### 3.1. In Vitro Screening for 3CL^pro^ and PL^pro^ Inhibition by Natural Compounds

A luminescent protease assay was carried out to verify the antiprotease activity of natural compounds. Recombinant SARS-CoV-2 3CL^pro^ and PL^pro^ were separately combined with luminogenic 3CL^pro^ substrate and natural compounds for 60 min at 4 °C. The reaction was then stopped by adding Luciferin Detection Reagent, and the luminescence was recorded on a plate-reading luminometer. The GC376 was used as a positive control. The results, expressed as relative luminescence units (RLU), are reported in Figure 2 and Figure 3, and the related IC_50_ is reported in Table 1.

The obtained IC_50_ values showed moderate inhibition of the enzymatic activity of 3CL^pro^ for L7OG and FA, whereas no enzymatic inhibition was observed for CY and RA (Table 1 and Figure 2). Similarly, L7OG and FA moderately inhibit papain-like protease PL^pro^ (Table 1 and Figure 3). As expected, GC376, used as a control, inhibited 3CL^pro^ in the low nanomolar range. Moreover, we report the percentage of enzymatic activity inhibition in panel B of Figure 2 and Figure 3, in which we screened the inhibition of viral proteases using 300 μM of compounds. The results demonstrated that L7OG and FA significantly reduced the % of SARS-CoV-2 3CL^pro^ and PL^pro^ activity. CY and RA, however, exhibited a non-significant reduction. For this reason, we focused on evaluating the activity of L7OG and FA against SARS-CoV-2 through pseudovirus assay and in silico analysis.

### 3.2. Pseudoviruses Entry Inhibition Screening of L7OG and FA

In the light of 3CL^pro^ and PL^pro^ inhibition results and cytotoxic assessment (see Appendix A for the CC_50_ values), L7OG and FA were tested to be effective against cellular transmembrane protease by pseudovirus technology employment. Two variants of pseudoviruses, bearing substantial mutations in the spike protein responsible for different entry mechanisms, were employed [47]. We produced a spike-based pseudovirus carrying a luciferase reporter gene through the coexpression of three plasmids, leading to the synthesis of MLV capsid proteins, spike envelope protein, and LTR-flanked luciferase, which permits the analysis of the spike-mediated entry into the cells [31]. Then, two different experimental conditions, cell pretreatment and pseudovirus pretreatment, were carried out to show if the interactions of natural compounds entail the cellular or viral entry machinery. In our experimental design, we hypothesized that by pretreating the cells with L7OG and FA, prior to transduction with both (alpha and omicron) pseudovirus variants, we would have verified whether the inhibitory action was on cellular proteases. Therefore, we incubated the VERO cells with L7OG or FA at no toxic concentration for 2 h, and then transduced them with α or o−-SARS-CoV-2 pseudoviruses for a further 2 h (cell-pretreatment protocol). The infectivity of pseudotyped viral particles was measured by luciferase assay 72 h post-transduction (Figure 4A). The selective entry inhibitors of Cathepsin (Calpeptin, 1 μM) and TMPRSS2 protease (Camostat, 5 μM) were used as positive controls. To investigate the direct inhibitory effect on pseudovirus by natural compounds, VERO cells were exposed to SARS-CoV-2 pseudoviruses pretreated with L7OG and FA (Figure 4B, pseudovirus-pretreatment protocol). The results, reported in Figure 4, show that both substances effectively block the entry of SARS-CoV-2 pseudoviruses into cells in the pseudovirus-pretreatment assay. We speculate that both compounds interfere with the SARS CoV-2 through a peculiar action on spike protein. FA blocked the entry of both α-SARS-CoV-2 and o−-SARS-CoV-2 pseudoviruses, and L7OG specifically blocked the entry of the α variant (Figure 5B). Our experimental approach allowed us to add further evaluations to previously reported results [18], identifying the specific SARS-CoV-2 entry inhibition mechanism mediated by L7OG.

### 3.3. Docking and Molecular Dynamics Simulations

The experimental in vitro studies were corroborated by molecular modeling analysis to identify and evaluate the receptor/ligand recognition’s fundamental molecular interactions with the two most active compounds FA and L7OG. To study the interactions with the spike proteins of the alpha and omicron variants, crystallographic structures of the RBD domains in complex with ACE2 and the entire closed spike protein were used.

For the L7OG/3CL^pro^ complex, a network of H-bonds was observed between the glucoside moiety of L7OG and the side chains of Thr24, Thr25, Thr26, and Ser46, while the flavone moiety result stabilized by H-bonds with Leu141, Gly143, Ser144, Cys145, Glu166, Arg188, and by π-π interaction with His4 and Met164 within the P4 pocket (Appendix A). To investigate the validity of docking, the best-docked pose of L7OG/3CL^Pro^ complex was subjected to MD simulation for 100 ns. The RMSD of the ligand shows a linear trend around 2 Å, showing peaks of fluctuations between 1.00 and 2.75 Å. It is worth noting how the starting structure undergoes an abrupt internal change at 60 ns (Appendix A). The complex’s poor stability during MD simulation validates the in vitro test results.

For the FA/3CL^pro^ complex, FA’s dihydropteridine ring forms two H-bonds with Glu166 at distances of 1.8 and 2.4 Å, stabilizing this part of the molecule inside the P1 pocket of the protein (Appendix A). The glutamic part of FA inserts into the catalytic site, forming additional H-bonds with the residues Thr26, His41, Phe140, and Asn119. Throughout the entire MD simulation, interactions involving Glu166 and ASN119 were maintained. The overall RMSD of the protein system reached equilibrium after 1 ns and protein–ligand complex stability after 10 ns, preserving the complex’s extensive hydrogen bond network.

The docking results are consistent with the in vitro experiments, revealing that L7OG and FA had a higher affinity for PL^pro^, with binding energies of −7.7 and −7.8 kcal/mol, respectively (Appendix A).

For L7OG/PL^pro^ complex, an extensive network of H-bonds was observed between the ligand and the active site of PL^pro^. The carboxyl group of the glucoside reaction forms a salt bridge with Arg166, while an H-bond is established with Glu167. Two H-bonds with Leu 162 and Tyr264 and π-π stacking interactions with Gly163, Pro248, and Tyr268 stabilize the flavone fraction (Appendix A). The stability of the complex during MD simulation shows larger fluctuations in the RMSD of the ligand (Appendix A).

For the FA/PL^pro^ complex, there is the formation of H-bonds with Lys157, Glu167, Arg166, Tyr264, and Thr301 inside the PL^pro^ active site. For the FA/PL^pro^ complex, a very slight variation was observed during the simulation. This slight instability could be due to the elongated structure of the ligand within the active site, which must change its folded conformation trying to fit into the binding cavity of PL^pro^. FA forms H-bonds with Lys157 and Thr301, salt bridges, and electrostatic interactions with Arg166, Tyr264, and Asp164 (Appendix A). The MD simulation does not show important fluctuations in the complex, keeping the main interactions stable (Appendix A).

Below, the docked poses of O7 and FA in the RBD domains of both variants will be analyzed.

The docked poses of FA and L7OG reveal that the adjacent α-RBD in the trimer shifts towards its neighbor, and the anchor residues Arg408 and Gln409 lock onto the compound head (Figure 5). Overall, this results in a compaction of the trimer architecture in the region formed by the three RBDs, producing a locked S-shaped structure. The greasy tube is flanked by a gate helix, with Arg395 and Gnl396 of SARS-CoV positioned 10 and 11 Å from the entrance, respectively. FA establishes an electrostatic interaction with Phe392 and π-π stacking with Leu387, while two π-alkyl interactions with Tyr365 and Tyr396 stabilize the central moiety of the ligand. Arg408 and Gln409 from a neighboring RBD form a fork, firmly holding the carboxyl tail of the bound FA through salt bridges and hydrogen bonds, which hinder the movement of the RBD and lock the RBDs in a completely “down” conformation, thus preventing their interaction with ACE2. L7OG mainly establishes π-π stacking and CH-π T-shaper interactions of the flavone moiety with Phe377, Phe374, and Tyr365, while two H-bonds with Arg408 and Thr415 are formed with the phenolic hydroxyls.

The docked pose of FA in complex with o−-RBD shows that the molecule is stabilized by an extensive network of H bonds with Arg393 (1.8 Å), Phe390 (2.9 Å), Glu37 (2.6 Å) and a salt bridge with Lys562 (1.9 Å), while the aromatic central part shows a double interaction on both planes of the molecule, π-π stacking at 5.1 Å with Phe40 and π-alkyl with Arg393 at 4.3 Å (Figure 6A).

A 100 ns MD simulation of the o−-RBD/ACE2 complex was performed to understand better the molecular dynamics resulting in a destabilization of the protein complex. The complex’s root mean square deviations (RMSDs) were analyzed using backbone atoms. The protein–ligand system showed an inconsistent RMSF during the MD simulation (Appendix A), probably due to the interaction of the NH_2_ group of FA with the residue Asp405 of ACE2, establishing an H-bond throughout the MD simulation at an average distance of 1.9 Å. Interestingly, after 100 ns of MD simulation, Lys562 and Phe390 of the oRBD domain maintained interactions with FA while losing H-bonding with Glu37. This may be attributed to the increased penetration of the ligand into the RBD/ACE2 protein interface (Figure 6).

The stability of the RBD-ACE2 complex was analyzed through the root mean square deviation (RMSD) of the atomic positions of the simulated protein structure from its native coordinates. The RMSD profile (shown in Appendix A) of the Cα atoms remained stable around average values of 1.9 ± 2.7 Å along the entire trajectory, indicating that the complex was stable throughout the simulation. The root mean square fluctuation (RMSF) helped to understand the flexibility of each amino acid residue during the simulation time. RMSF analysis showed significant fluctuations between residues 125–145, 357–402, and 507–528, with RMSF values from 1.5 Å to 7.1 Å (Figure 6D).

The free energy of binding and the various energetic components of protein–protein interactions inside the o−-RBD/ACE2 complex, calculated by Foldx, present unequivocal data on the protein complex’s stability exerted by the FA. The free energy is determined by FoldX 4.0 software employing the plugin implemented in YASARA. After 100 ns of MD simulation, the binding energy of the o−-RBD/ACE2 complex goes from −10.8 kcal/mol to −4.1 kcal/mol (Figure 6E), suggesting a drastic loss of interaction in the interface between the two proteins. Indeed, a constant tendency towards loosening the protein complex’s interactions is evident.

## 4. Discussion

The paper explores the potential antiviral properties of L7OG and FA. The study initially evaluated the inhibitory activity of L7OG and FA against viral proteases 3CL^pro^ and PL^pro^, finding moderate activity, with detailed molecular modeling studies providing insights into their interactions. While the L7OG/PL^pro^ complex showed fluctuations during MD simulation, FA demonstrated better stability. Moreover, the study investigated the effectiveness of L7OG and FA against cellular transmembrane proteases using pseudoviruses with mutations in the spike protein associated with different entry mechanisms [47]. Despite the omicron-variant mutations favoring a cathepsin-dependent endosomal entry route [48,49], neither L7OG nor FA inhibited the entry of pseudoviruses in the experimental design, aligning with weak activity observed in modeling studies (Figure 5A). In a subsequent experiment, both compounds were found to interfere with SARS-CoV-2 by acting on the spike protein (Figure 5B). FA blocked the entry of both α and ο pseudoviruses, while L7OG specifically inhibited the entry of the α variant. The study’s in vitro results suggest that FA blocks SARS-CoV-2 entry by binding to the spike protein, confirmed by computational docking analysis. Notably, FA exhibited powerful binding energies for the α variant’s receptor-binding domain (RBD) but lower affinity for the ο variant due to unique mutations in the hydrophobic pocket of the RBD, affecting the interaction with FA and potentially altering the RBD/ACE2 binding. Our results overlap with the numerous studies conducted on the antiviral activity of L7OG and FA against SARS-CoV-2. In fact, in silico and molecular docking studies reported the potential association between FA and SARS-CoV-2 [21]. The study by Kaur et al. recorded that FA is found to bind to furin-protease and the spike protein–human ACE2 interface of SARS-CoV-2 [50]. In the study by Chen et al., molecular docking showed that FA could act on SARS-CoV-2 nucleocapsid phosphoprotein (SARS-CoV-2 N) [25]). Ugurel et al. reported that FA was among the most potent drugs inhibiting wild-type and mutant SARS-CoV-2 helicase [51]. However, all mentioned studies have a consensus that the impact of FA on viruses is not fully known and requires further investigation. Solely, Zhang et al. and Škrbić, R et al. individuated a potential biological mechanism that justifies FA’s effect on SARS-CoV-2. In the first case, it was reported that folic acid affects the expression of ACE2 by regulating methylation in the promoter region of ACE2 [52]. Second, it was hypothesized that folic acid interacted with the NRP-1 receptor-binding domain, which is a co-receptor of SARS-CoV-2, thereby preventing endocytosis [53]. Moreover, regarding L7OG, an extensive bibliography of in silico analysis has demonstrated the effect against SARS-CoV-2 [54,55,56,57,58]. In a few papers, such as ref. [57,59], the molecular mechanisms responsible for the inhibitory effect were determined. For example, Dissook and coauthors showed that in a concentration-dependent manner, luteolin effectively inhibits spike S1-induced main inflammatory mediators, including IL-6, IL-1β, and IL-18. The results of our study are consistent with findings previously reported by Zhu et al., where luteolin was shown to suppress the entry of different types of SARS-CoV-2 S-protein-pseudoviruses into ACE2 overexpressing 293T cells [18]. The agreement between our study’s results and those of Zhu et al. provides additional support for the inhibitory effects of the studied compound on the entry of SARS-CoV-2 pseudoviruses, reinforcing the potential significance of these findings in the context of antiviral research. Unlike previously reported papers, we reported an inhibitory activity of both compounds on the entry for two variants of SARS-CoV-2 through pseudovirus technology and used an in silico approach to complement and support the findings from in vitro studies. Thus, our study is positioned as innovative and impactful due to its exploration of folic acid’s inhibitory effects on viral variants, for which data are currently unavailable. The research can potentially fill a crucial knowledge gap and contribute to the broader understanding of antiviral properties, especially in emerging viral variants.

## 5. Conclusions

In this specific study, we comprehensively evaluated the inhibitory activity of two natural compounds, L7OG and FA, against key viral proteases (3CL^pro^ and PL^pro^) associated with SARS-CoV-2. Despite the compounds exhibiting only moderate activity, our molecular modeling studies provided detailed insights into their interactions. The novelty of our work lies in the specific focus on these compounds and their interactions with viral proteins, shedding light on potential mechanisms of action. Unlike previous papers, we reported an inhibitory activity of FA and L7OG on the entry for two variants of SARS-CoV-2 through pseudovirus technology and used an in silico approach to complement and support the findings from in vitro studies. Molecular modeling studies demonstrated that FA and L7OG can bind to the deep hydrophobic pocket of the RBD domain of the alpha variant located on top of the SARS spike protein trimer-CoV-2. Possibly, both bound compounds mediate strong interactions between the “down” RBDs, and lock the majority of S trimers in an “all down” and ACE2-inaccessible inhibitory conformation.

This work allowed us to identify the binding sites of FA and L7OG within the RBD domain in the alpha and omicron variants, demonstrating how FA is active on both variants by acting in different RBD domain regions.

## Figures and Tables

**Figure 1 biomolecules-14-00043-f001:**
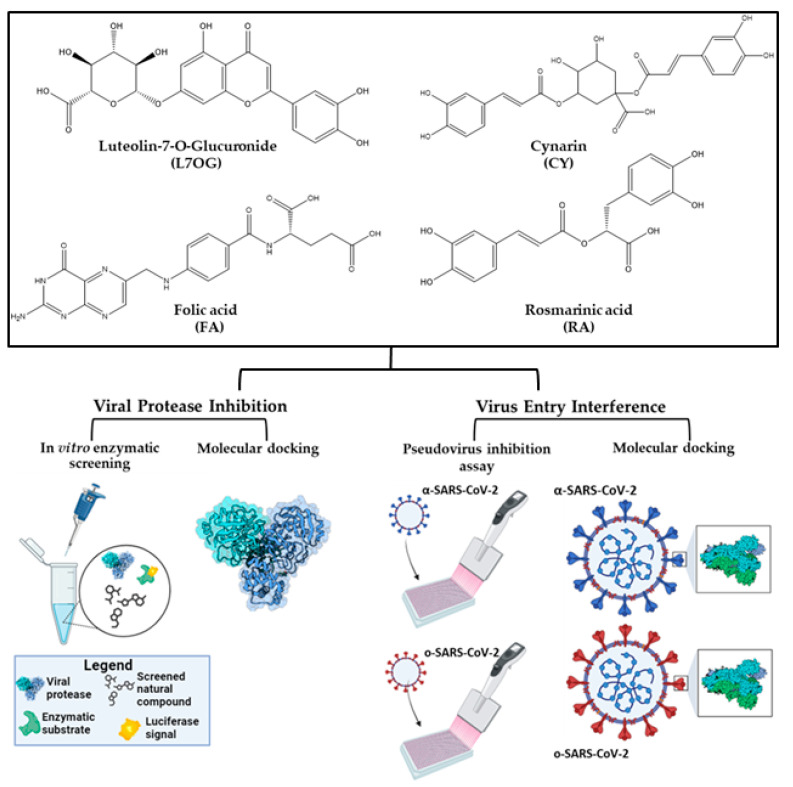
Workflow of the integrated in silico and in vitro approach.

**Figure 2 biomolecules-14-00043-f002:**
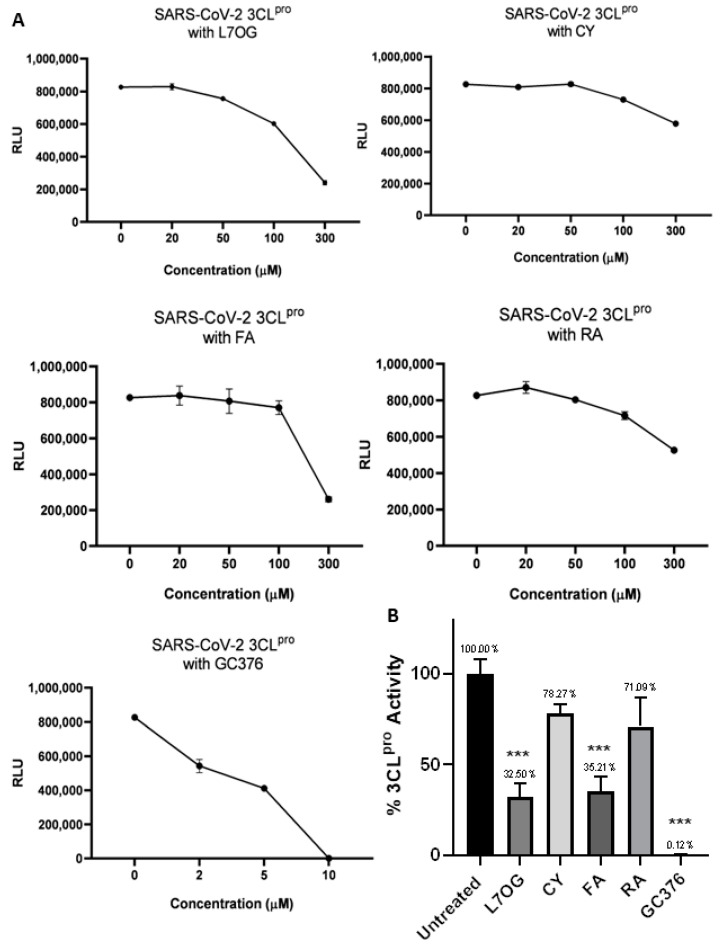
In vitro inhibition of 3CL^pro^ activity. Purified recombinant SARS-CoV-2 3CL^pro^ was combined with L7OG, RA, CY, and FA, in the concentration range of 20–300 μM, and added to 3CL^pro^ substrate solution for 60 min at 37 °C. GC376 (2 μM, 5 μM, and 10 μM) was used as positive 3CL^pro^ inhibitor control. The reaction was stopped by adding Luciferin Detection Reagent, and the luminescence was recorded on a plate-reading luminometer. The results were expressed as relative luminescence units (RLU) (**A**) and percentage of enzymatic activity inhibition (**B**). *** *p* < 0.001.

**Figure 3 biomolecules-14-00043-f003:**
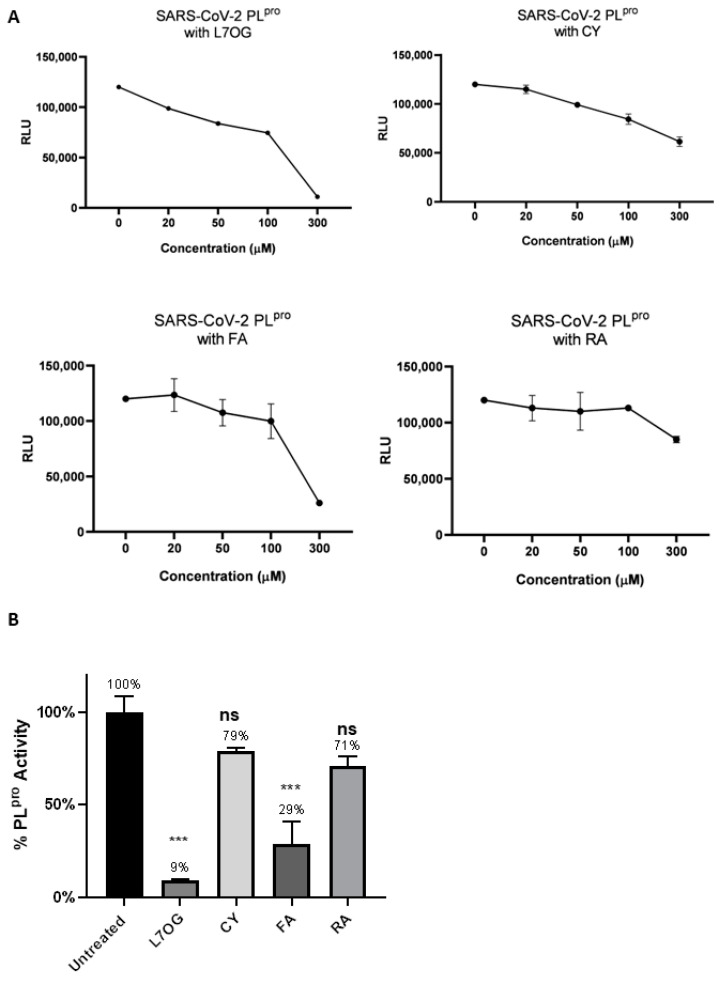
In vitro inhibition of PL^pro^ activity. Purified recombinant SARS-CoV-2 PL^pro^ was combined with L7OG, CY, FA, and RA in the 20–300 μM concentration range and added to 3CL^pro^ substrate solution for 60 min at 37 °C. The reaction was stopped by adding Luciferin Detection Reagent, and the luminescence was recorded on a plate-reading luminometer. The results were expressed as relative luminescence units (RLU) (**A**) and percentage of enzymatic activity inhibition (**B**). *** *p* < 0.001.

**Figure 4 biomolecules-14-00043-f004:**
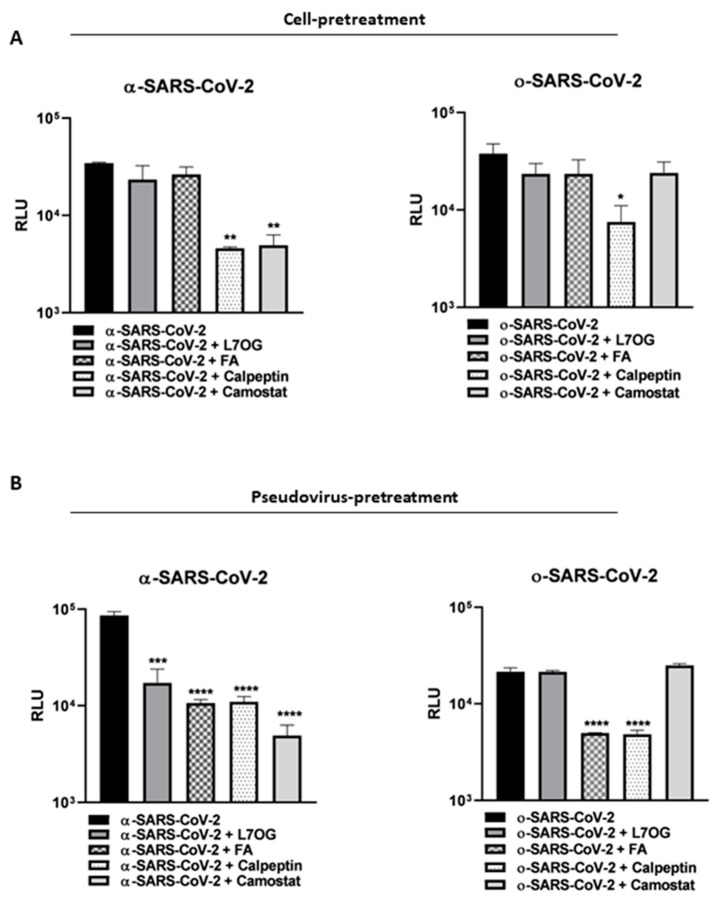
SARS-CoV-2 spike pseudotyped particle entry inhibition assay. (**A**) Vero cells were pretreated with L7OG (5 μM), FA (20 μM) and Camostat (inhibitor of the serine protease TMPRSS2) (5 μM) and Calpeptin (CatL inhibitor) (1 μM) for 2 h at 37 °C and then transduced with α-SARS-CoV-2 and o−-SARS-CoV-2 pseudoviruses for 2 h with gentle agitation. (**B**) The α-SARS-CoV-2 and o−-SARS-CoV-2 pseudoviruses were pretreated with L7OG (5 μM), FA (20 μM), and Camostat (inhibitor of the serine protease TMPRSS2) (5 μM), and Calpeptin (CatL inhibitor) (1 μM) for 1 h at 4 °C and then transduced on VERO cells for 2 h. The luciferase activity was revealed 72 h post-transduction. The results are the means ± SD of triplicate analyses and are expressed as relative fluorescence units (RLU). * *p* < 0.05, ** *p* < 0.01, *** *p* < 0.001, **** *p* < 0.0001.

**Figure 5 biomolecules-14-00043-f005:**
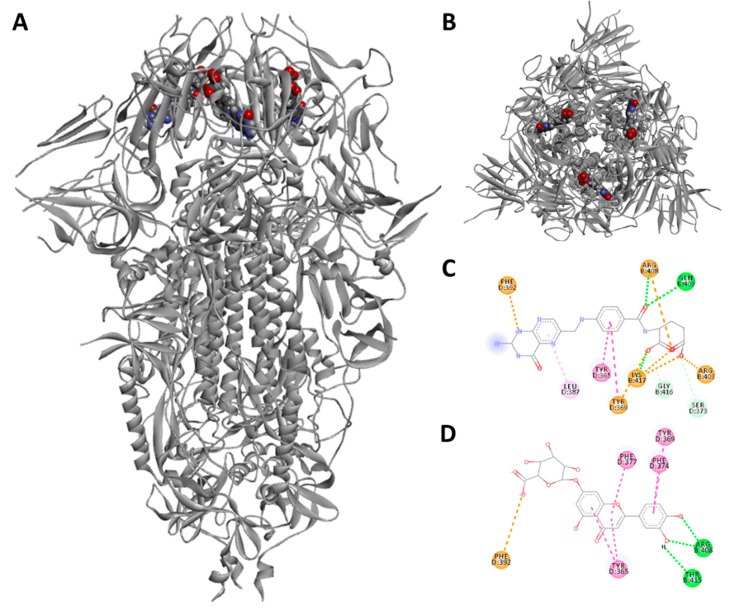
(**A**) Spike protein (alpha variant) in the closed conformation bound to three molecules of FA; (**A**,**B**) its view from above; (**C**) 2D interactions of FA; (**D**) L7OG docked within the hydrophobic pocket of the RBD domain.

**Figure 6 biomolecules-14-00043-f006:**
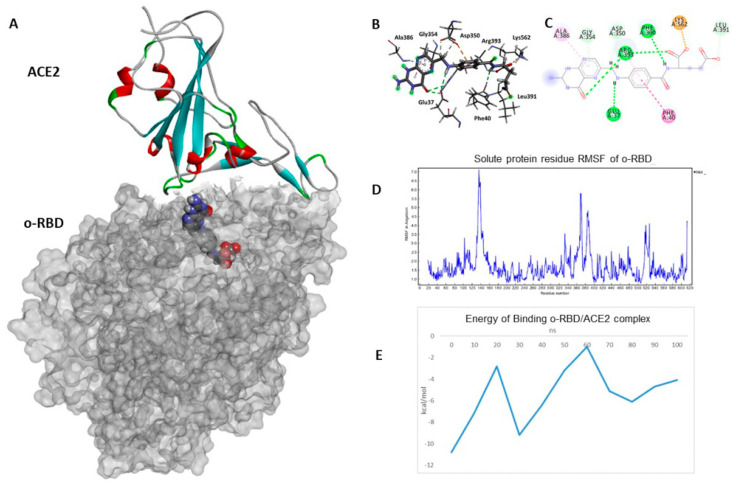
(**A**) Hydrophobic surface o−-RBD domain in gray and human ACE2 in complex with FA within a sub-pocket of o−-RBD; (**B**) interaction profile of the docked poses of FA; (**C**) and 2D diagram interaction profile; (**D**) RMSF of the atoms constituting the o−-RBD domain; (**E**) energy of binding variation of the o−-RBD/ACE2 in complex with FA during MD simulation.

**Table 1 biomolecules-14-00043-t001:** IC_50_ values (μM) for inhibiting SARS-CoV-2 3CL^pro^ and PL^pro^ by natural compounds and the GC376 used as a positive control.

Compound	3CL^pro^	PL^pro^
L7OG	170.0	152.7
CY	573.9	150.0
FA	232.0	183.0
RA	394.0	338.2
GC376	5.5	—

## Data Availability

The data presented in this study are available on request from the corresponding author.

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
