# Peer review of "An Integrated In Silico and In Vitro Approach for the Identification of Natural Products Active against SARS-CoV-2"

_biomolecules, 2023, doi:10.3390/biom14010043_

Round 1

Reviewer 1 Report

Comments and Suggestions for Authors
  1. Is there any previous study showing the effect of FA and L7OG on Coronaviruses? If so, what was their observation?
  2. What is the mechanism of action for L7OG-induced inhibitory entry activity of SARS-CoV-2?
  3. Since SARS-CoV-2 is a respiratory virus, using primary respiratory epithelial cells from Humans in addition to the Vero cells used here may give a broader perspective on the study.

Author Response

Response to Reviewer 1:

We sincerely thank the Reviewer for the comments, which were of great help in revising the manuscript. Accordingly, the revised manuscript has been systematically improved. Below is a point-by-point description that includes the original Reviewer’s comments in boldface and the responses in red regular typeface. We specify that we resubmit the new version of the manuscript with several corrections compared to the previous version.

  1. Is there any previous study showing the effect of FA and L7OG on Coronaviruses? If so, what was their observation?

We thank the Reviewer for the observation. We reported the studies regarding the activity of FA and L70G on SARS-CoV-2 in the discussion section as follows:

 “ The paper explores the potential antiviral properties of L7OG and FA. The study initially evaluated the inhibitory activity of L7OG and FA against viral proteases 3CLPro and PLPro, finding moderate activity, with detailed molecular modeling studies providing insights into their interactions. While the L7OG/PLPro complex showed fluctuations during MD simulation, FA demonstrated better stability. Besides, the study investigated the effectiveness of L7OG and FA against cellular transmembrane proteases using pseudoviruses with mutations in the spike protein associated with different entry mechanisms [47]. Despite the omicron variant mutations favoring a cathepsin-dependent endosomal entry route [48,49], neither L7OG nor FA inhibited the entry of pseudoviruses in the experimental design, aligning with weak activity observed in modeling studies (Figure 5A). In a subsequent experiment, both compounds were found to interfere with SARS-CoV-2 by acting on the spike protein (Figure 5B). FA blocked the entry of both α and ο pseudoviruses, while L7OG specifically inhibited the entry of the α variant. The study’s in vitro results suggest that FA blocks SARS-CoV-2 entry by binding to the spike protein, confirmed by computational docking analysis. Notably, FA exhibited powerful binding energies for the α variant’s receptor-binding domain (RBD) but lower affinity for the ο variant due to unique mutations in the hydrophobic pocket of the RBD, affecting the interaction with FA and potentially altering the RBD/ACE2 binding. Our results overlap with the numerous studies conducted on the antiviral activity of L7OG and FA against SARS-CoV-2. In fact, in silico and molecular docking studies reported the potential association between FA and SARS-CoV-2 [21]. The study by Kaur et al. recorded that FA is found to bind to furin-protease and the spike protein–human ACE2 interface of SARS-CoV-2 [50]. In the study by Chen et al., molecular docking showed that FA could act on SARS-CoV-2 nucleocapsid phosphoprotein (SARS-CoV-2 N) [51].) Ugurel et al. reported that FA was among the most potent drugs inhibiting wild-type and mutant SARS-CoV-2 helicase [52]. However, all mentioned studies have a consensus that the impact of FA on viruses is not fully known and requires further investigation. Solely, Zhang et al. and Škrbić, R et al. individuated a potential biological mechanism that justifies FA’s effect on SARS-CoV-2. In the first case, it was reported that folic acid affects the expression of ACE2 by regulating methylation in the promoter region of ACE2 [53]. Second, it was hypothesized that folic acid interacted with the NRP-1 receptor-binding domain, which is a co-receptor of SARS-CoV-2, thereby preventing endocytosis [54]. Moreover, regarding L7OG, an extensive bibliography of in silico analysis has demonstrated the effect against SARS-CoV-2 [55–59]. In a few papers, [60] the molecular mechanisms responsible for the inhibitory effect [58] were determined. For example, Dissook and coauthors showed that in a concentration-dependent manner, luteolin effectively inhibits Spike S1-induced main inflammatory mediators, including IL-6, IL-1β, and IL-18. The results of our study are consistent with findings previously reported by Zhu et al., where luteolin was shown to suppress the entry of different types of SARS-CoV-2 S-protein-pseudoviruses into ACE2 overexpressing 293T cells [61]. The agreement between our study’s results and those of Zhu et al. provides additional support for the inhibitory effects of the studied compound on the entry of SARS-CoV-2 pseudoviruses, reinforcing the potential significance of these findings in the context of antiviral research. Unlike previously reported papers, we reported an inhibitory activity of both compounds on the entry for two variants of SARS-CoV-2 through pseudovirus technology and used an in-silico approach to complement and support the findings from in vitro studies. Thus, our study is positioned as innovative and impactful due to its exploration of folic acid’s inhibitory effects on viral variants, for which data is currently unavailable. The research can potentially fill a crucial knowledge gap and contribute to the broader understanding of antiviral properties, especially in emerging viral variants.”

We also re-edited the references accordingly.

  1. What is the mechanism of action for L7OG-induced inhibitory entry activity of SARS-CoV-2?

We thank the Reviewer for the query. The results reported in our study show that L7OG specifically blocked the entry of the α-SARS-CoV-2 variant, inhibiting the S-protein RBD-ACE2 interaction. Molecular modeling studies support in vitro experimental data, indicating that L7OG can bind to the deep hydrophobic pocket of the receptor-binding domain (RBD) of the alpha variant of the SARS-CoV-2 spike protein. This binding occurs on top of the SARS-CoV-2 spike protein trimer. Additionally, it was suggested that the binding may mediate strong interactions between the “down” RBDs, locking the majority of spike trimers in an “all down” conformation inaccessible to ACE2, thereby inhibiting the virus entry.

  1. Since SARS-CoV-2 is a respiratory virus, using primary respiratory epithelial cells from Humans in addition to the Vero cells used here may give a broader perspective on the study.

We understand the Reviewer’s concern; however, since 2003, Vero cells have been used extensively for SARS-CoV and recently for SARS-CoV-2 due to the common affinity of the two viruses for the angiotensin-converting enzyme 2 (ACE2) receptor, and because this receptor is abundantly expressed in Vero cells, indeed after the 2020 outbreak Vero cells were used for research in cell-culture-based infection models by many laboratories. Nonetheless, in the future, we will evaluate the possibility of using a lung cell model to improve the medicinal translational relevance of our new results. 

Reviewer 2 Report

Comments and Suggestions for Authors

The paper submitted for review is interesting, it concerns quite hot topic, but many of its elements require correction before possible publication:

1) it is not clear why a set of such 4 compounds was chosen, have they anything common?

2) a significant part of the conclusions section should be included in the introduction or discussion because these are not the authors' own results

3) conclusions should be written again presenting what was done in this specific work, generalities should be avoided, the novelty should be clearly stated

4) the introduction is very extensive but little related to the subject of research, e.g. no justification for choosing these and not other compounds, e.g., luteolin, cynarin, folic acid and rosmarinic acid were the subject of intensive research for SARS (55, 2, 10 and 8 papers respectively) but the authors do not mention it; the type of the SARS virus (omicron, alpha etc.) is of marginal importance because they all contain the same spike and the mode of binding is the same.

5) what algorithm was used to calculate binding energies? there are many different ones, which one specifically?

6) conclusions regarding the action of luteolin should be compared with the results of Molecules' work. 2023 Mar; 28(5): 2294

7) the authors report the results but do not discuss their meaning in details, nor do they compare their results with the others.

Comments on the Quality of English Language

The English language needs slight improvement, there are strange phrases in the paper, e.g. "the MD of the ligand shows notable fluctuations during MD Simulation".

Author Response

We sincerely thank the Reviewer for the comments, which were of great help in revising the manuscript. Accordingly, the revised manuscript has been systematically improved. Below is a point-by-point description that includes the original Reviewer’s comments in boldface and the responses in red regular typeface. We specify that we resubmit the new version of the manuscript with several corrections compared to the previous version.

The paper submitted for review is interesting, it concerns quite hot topic, but many of its elements require correction before possible publication:

  1. it is not clear why a set of such 4 compounds was chosen, have they anything common?

A thorough review of the literature guided our selection of L7OG, CY, FA, and RA, which have been reported to demonstrate activity against the viral protease of SARS-CoV-2. In particular, RA and CY, both polyphenols derived from caffeic acid, have been the subject of significant debate in the existing literature regarding their efficacy against SARS-CoV-2. In contrast, the literature-supported efficacy of FA and L7OG against SARS-CoV-2 was reaffirmed through these studies.

Despite the multitude of in silico tests detailed in the text, the precise mechanism of inhibition of these compounds remains elusive. Consequently, these substances were chosen to explore the wealth of non-exhaustively documented studies present in the literature, trying to shed light on their potential antiviral mechanisms.

  1. a significant part of the conclusions section should be included in the introduction or discussion because these are not the authors' own results.

We are grateful to the Reviewer for this suggestion. We edited the Discussion paragraph as per the Reviewer’s suggestions.

  1. conclusions should be written again presenting what was done in this specific work, generalities should be avoided, the novelty should be clearly stated.

We are grateful to the Reviewer for this suggestion. We edited the Discussion paragraph as per the Reviewer’s suggestions.

  1. the introduction is very extensive but little related to the subject of research, e.g. no justification for choosing these and not other compounds, e.g., luteolin, cynarin, folic acid and rosmarinic acid were the subject of intensive research for SARS (55, 2, 10 and 8 papers respectively) but the authors do not mention it; the type of the SARS virus (omicron, alpha etc.) is of marginal importance because they all contain the same spike and the mode of binding is the same.

We thank the Reviewer for the observation. For better readability of our manuscript, we added panel B in Figures 2 and 3. We re-edited the introduction:

“From an initial screening of the inhibition of viral proteases, performed using 300 µM of compounds, we obtained that L7OG and FA significantly reduced the % of SARS-CoV-2 3CLpro and PLpro activity. CY and RA, however, exhibited a non-significant reduction. For this reason, we focused on evaluating the activity of L7OG and FA against SARS-CoV-2 through pseudoviruses assay and in silico analysis.”

Besides we also edited section 3.1 as follows:

“Besides, we report the percentage of enzymatic activity inhibition in panel B of Figures 2 and 3, in which we screened the inhibition of viral proteases using 300 µM of compounds. The results demonstrated that L7OG and FA significantly reduced the % of SARS-CoV-2 3CLpro and PLpro activity. CY and RA, however, exhibited a non-significant reduction. For this reason, we focused on evaluating the activity of L7OG and FA against SARS-CoV-2 through pseudoviruses assay and in silico analysis.”

  1. what algorithm was used to calculate binding energies? there are many different ones, which one specifically?

AutoDock 4.2.6 uses a semi-empirical free energy force field to evaluate conformations during docking simulations. The force field was parameterized using many protein-inhibitor complexes for which both structure and inhibition constants, or Ki, are known. The weighting constants W have been optimized to calibrate the empirical free energy based on a set of experimentally determined binding constants. The first term is a typical 6/12 potential for dispersion/repulsion interactions. The parameters are based on the Amber force field. The second term is a directional H-bond term based on a 10/12 potential. The parameters C and D are assigned to give a maximal well depth of 5 kcal/mol at 1.9Å for hydrogen bonds with oxygen and nitrogen and 1 kcal/mol at 2.5Å for hydrogen bonds with sulfur. The function E(t) provides directionality based on the angle t from ideal H-bonding geometry. The third term is a screened Coulomb potential for electrostatics. The final term is a desolvation potential based on the volume of atoms surrounding a given atom and sheltering it from solvent, weighted by a solvation parameter and an exponential term with distance-weighting factor σ = 3.5Å.

  1. conclusions regarding the action of luteolin should be compared with the results of Molecules' work. 2023 Mar; 28(5): 2294

We thank the Reviewer for the observations. However, the proposed article reported the role of folic acid as a potential inhibitor of SARS-CoV-2 entry, and it is not related to luteolin, as mentioned. In this study, the authors reported that folic acid could prevent the SARS-CoV-2 virus' entry into host cells, interacting with the NRP-1 receptor-binding domain, which serves as a co-receptor of SARS-CoV-2, thereby preventing the endocytosis. We added the reference number 54 and re-edited the texts accordingly.

  1. the authors report the results but do not discuss their meaning in details, nor do they compare their results with the others.

We thank the Reviewer for the observations. We re-edited the texts as shown in the manuscript.

  1. Comments on the Quality of English Language

The English language needs slight improvement, there are strange phrases in the paper, e.g. "the MD of the ligand shows notable fluctuations during MD Simulation".

We thank the Reviewer for the observations. We re-edited the texts as shown in the manuscript.

Round 2

Reviewer 2 Report

Comments and Suggestions for Authors

The amendments are satisfactory.